# COVID-19 and Pulmonary Thrombosis—An Unresolved Clinical Puzzle: A Single-Center Cohort Study

**DOI:** 10.3390/jcm11237049

**Published:** 2022-11-29

**Authors:** Loris Močibob, Frano Šušak, Maja Šitum, Klaudija Višković, Neven Papić, Adriana Vince

**Affiliations:** 1Department for Viral Hepatitis, University Hospital for Infectious Diseases Zagreb, Mirogojska 8, 10000 Zagreb, Croatia; 2Department of Infectious Diseases, School of Medicine, University of Zagreb, 10000 Zagreb, Croatia; 3Department for Radiology and Ultrasound Diagnostics, University Hospital for Infectious Diseases Zagreb, Mirogojska 8, 10000 Zagreb, Croatia; 4Faculty of Health Studies, University of Rijeka, 51000 Rijeka, Croatia

**Keywords:** pulmonary thrombosis, COVID-19, D-dimer, pulmonary angiography, CTPA, venous thromboembolism

## Abstract

Pulmonary thrombosis (PT) is a frequent complication of COVID-19. However, the risk factors, predictive scores, and precise diagnostic guidelines on indications for CT pulmonary angiography (CTPA) are still lacking. This study aimed to analyze the clinical and laboratory characteristics associated with PT in patients with COVID-19. We conducted a cohort study of consecutively hospitalized adult patients with COVID-19 who underwent CTPA at the University Hospital for Infectious Diseases in Zagreb, Croatia between 1 April and 31 December 2021. Of 2078 hospitalized patients, 575 (27.6%) underwent CTPA. PT was diagnosed in 178 (30.9%) patients (69.6% males, median age of 61, IQR 50–69 years). The PT group had a higher CRP, LDH, D-dimer, platelets, and CHOD score. PT was more frequent in patients requiring ≥15 L O_2_/min (25.0% vs. 39.7%). In multivariable analysis, only D-dimer ≥ 1.0 mg/L (OR 1.78, 95%CI 1.12–2.75) and O_2_ ≥ 15 L (OR 1.89, 95%CI 1.26–2.84) were associated with PT. PT was not associated with in-hospital mortality. In conclusion, our data confirmed a high incidence of PT in hospitalized patients with COVID-19, however, no correlation with traditional risk factors and mortality was found. CTPA should be performed in patients requiring high-flow supplemental oxygen or those with increased D-dimer levels.

## 1. Introduction

Venous thromboembolism (VTE) is a frequent complication of COVID-19, with an estimated incidence ranging from 4.8% to 85% [1]. Of all VTE events, the risk of pulmonary thrombosis (PT) is the highest, especially in the acute phase of the disease, [2]. However, traditional risk factors for PT, such as previous VTE, history of cancer, or chronic heart failure usually have no association with PT in COVID-19 [3]. PT in patients with COVID-19 is thought to be of different pathogenesis than the usual embolic phenomenon of pulmonary embolism. In situ PT may be caused by direct endothelial injury due to local inflammation and hypercoagulable state in combination with occult hyperfibrinolysis, as was recently reported [4,5,6]. Suspecting a PT in a COVID-19 patient is often difficult since signs and symptoms of severe COVID-19 pneumonia cannot clearly be distinguished from PT and clinical probability scores, such as the Wells score, are unreliable [7]. While D-dimer are typically increased in patients with COVID-19 and their levels are even higher in patients with PT, the threshold to differentiate COVID-19 patients with and without PT has not been widely accepted [8]. Notably, there is a lack of precise diagnostic guidelines for computed tomography pulmonary angiography (CTPA) [9,10]. Imaging studies show that thrombotic lesions are mostly located in the peripheral rather than in the main pulmonary arteries, which suggests a different pathophysiological mechanism [11]. The clinical significance of subsegmental PT as an incidental finding is unknown [12,13]. In hospitalized patients, low-molecular weight heparin is recommended in doses based on clinical severity but not on CTPA or PT findings [14].

Here, we performed a cohort study to analyze the clinical and laboratory characteristics associated with PT in patients with COVID-19.

## 2. Materials and Methods

### 2.1. Study Design, Patients and Data Collection

In this retrospective analysis of prospectively collected data (part of the COVID-FAT study, ClinicalTrials.gov Identifier: NCT04982328), we included consecutively hospitalized adult patients with COVID-19 who underwent CTPA at the University Hospital for Infectious Diseases (UHID) in Zagreb, Croatia from 1 April 2021 to 31 December 2021. According to the ECDC database the predominating SARS-CoV-2 variant at the time was Delta (B.1.617.2 and AY lineages [15]. In all patients, COVID-19 diagnosis was confirmed by real-time polymerase chain reaction (RT-PCR) of a nasopharyngeal swab. Parameters included in the analysis were demographic data, comorbidities, routine laboratory tests, and clinical findings upon hospital admission. CHOD score, an acronym of CRP concentration + Heart rate + Oxygen saturation + D-Dimer levels, was calculated for each patient, as previously reported [16].

Patients were treated according to the current standard of care (including remdesivir, dexamethasone, low molecular weight heparin and tocilizumab and baricitinib). As per hospital protocol and international guidelines at that time, direct oral anticoagulants and warfarin were converted to LMWH dosed according to body weight and renal function. Data on clinical course, including oxygen requirements, invasive and non-invasive ventilation and complication rates, ICU admission and mortality were collected. Indication for CTPA was left at the physicians’ discretion since there was no hospital protocol at that time in UHID.

Patients were divided into two groups regarding positive (PT group) or negative (non-PT group) diagnosis of PT on CTPA. The study was conducted in accordance with the Declaration of Helsinki and approved by the Ethical Committee of the University Hospital for Infectious Diseases in Zagreb, Croatia (code 01–673-4–2021).

### 2.2. CTPA Protocol

FCT Speedia HD (Fujifilm corporation, Tokyo, Japan, 2017.) 64-detector MDCT scanner was used to acquire images of the thorax in cranio-caudal direction. For intravenous access an 18–20-Gauge catheter was introduced in a peripheral vein (antecubital vein was preferred). The chest field of view (FOV) comprised the widest rib-to-rib distance acquired during breath-hold after inspiration. Images were acquired with a standard algorithm and viewed with Hitachi, Ltd. Whole Body X-ray CT System Supria Software (System Software Version: V2.25) (Copyright Hitachi, Ltd.2017). Intravenous contrast media (Ultravis 370-Iopromidum, Bayer) was administrated with the help of a power injector (Medrad Stellant, Bayrt, Pennsylvania, USA) at a rate of 4 mL/s. Bolus tracking method was used with the cursor placed in the main pulmonary artery at a threshold of 120 HU to trigger the scanning. Acquired CT images were viewed on Fujifilm Synapse 3D workstation (Fujifilm corporation7–3, Akasaka 9-chome, Minato-ku, Tokyo 107–0052, Japan, 2017) in the axial, coronal and sagittal planes. For analysis of pulmonary arteries in postcontrast images, they were displayed in greyscale in a pulmonary embolism-specific setting (window width/level (HU) 700/100). For each lung the main, lobar, segmental and subsegmental arteries were analyzed. Direct axial sections and multiplanar reformatted images through the coronal and sagittal axes of vessels were used for differentiation between true pulmonary embolism and a variety of artefacts. For each participant with detected PE, a minimum of 10 consecutive images saved in DICOM format, in axial, coronal and sagittal plane were captured from Picture Archiving and Communication System (PACS), anonymized, and saved as JPG. At each image, an experienced senior consultant in radiology annotated lumen of occluded vessel or peripheral intraluminal filling defect using Microsoft Paint Software. A total of three experienced senior radiology consultants were included in image evaluation with one CTPA scan being analyzed by one radiologist.

### 2.3. Statistical Analysis

Clinical characteristics, laboratory tests and demographic data were evaluated and descriptively presented. Fisher’s exact test and the Mann–Whitney U test were used to compare the two groups. All tests were two-tailed; *p*-value < 0.05 was considered statistically significant. Risk factors associated with PT and mortality were investigated using a univariate and subsequently multivariable logistic regression analysis. The strength of association was expressed as odds ratio (OR) and its corresponding 95%CI. Statistical analyses were performed using GraphPad Prism Software version 9.4.1. (San Diego, CA, USA).

## 3. Results

### 3.1. Baseline Patients’ Characteristics

In the period studied, 2078 patients were hospitalized due to COVID-19. Of those, 575 (27.6%) underwent CTPA; 382 (66.4%) males with median age of 62 (IQR 51–69) years. Pulmonary thrombosis was diagnosed in 178 (30.9%) patients (124 male, median age 61, IQR 50–69 years). The median time from symptom onset to CTPA was 16 (IQR 13–20) days in both groups. Subsegmental PT was found in 64 (35.9%) patients, segmental in 84 (47.2%) patients. Lobar and central PT were diagnosed in 22 (12.4%) and 8 (4.5%) patients, respectively.

The baseline characteristics of patients with pulmonary thrombosis and patients without pulmonary thrombosis are shown in Table 1. Patients in PT group had significantly higher CRP and LDH levels, D-dimer values, platelet count, international normalized ratio (INR), and calculated CHOD score. There was no difference in age, sex, comorbidities, or other analyzed laboratory parameters, including the platelet-to-lymphocyte and neutrophil-to-lymphocyte ratio. In addition, there was no difference in chronic anticoagulation or antiaggregation (acetylsalicylic acid *n* = 56; warfarin *n* = 7; rivaroxaban *n* = 7; apixaban 3; fraxiparine *n* = 1).

During the hospital stay, the majority of patients were treated with dexamethasone (89.4%), remdesivir was applied in 32.7% of patients, while tocilizumab and baricitinib were used in 3.8% and 1.7% of patients, respectively. All patients received low molecular weight heparin (LMWH) prophylactic doses, however patients with diagnosed PT received weight-based therapeutic dose of LMHW.

Overall, 65 patients were admitted to the ICU (28, 15.7% in PT and 37, 9.3% in non-PT group). There were no differences in mortality; 13 (8.1%) with PT and 27 (7.1%) without PT died during hospitalization. The patients with PT had a longer duration of hospitalization (14, IQR 10–21 vs. 13, IQR 9–17 days, *p* = 0.015).

### 3.2. Association of Pulmonary Thrombosis with COVID-19 Severity

Next, we examined the association of PT with a level of respiratory support. Patients were divided into subgroups according to oxygen supplementation during hospitalization, as presented in Table 2 and Figure 1. The highest frequency of PT was observed among patients requiring more than 15 L of O_2_/min (44% in patients on 15–25 L of O_2_ on mask with reservoir, 37% in HFNC and 46% on NIV, 34% in IMV or ECMO).

Of the 237 patients requiring ≥15 L O_2_/min (including high-flow nasal cannula, non-invasive or invasive mechanical ventilation or needed extracorporeal membrane oxygenation), 94 (39.7%) were diagnosed with PT, which was significantly higher than in low-flow supplemental oxygen (<15 L/min) group (*n* = 86, 25.0%, *p* = 0.0002).

Furthermore, subsegmental peripheral PT was more commonly diagnosed in the low-flow supplemental oxygen group (35, 50.7%) as compared to 30 (31.9%) patients in the high-flow supplemental oxygen group.

### 3.3. Factors Associated with Pulmonary Thrombosis

We performed a univariable and multivariable analysis to identify factors associated with PT, as presented in Table 3. On univariable analysis, CHOD score ≥ 5, platelet count ≥ 185 × 10^9^/L, CRP ≥ 85 mg/L, D-dimer ≥ 1.0 mg/L, oxygen requirement ≥ 15 L/min, and ICU admission were associated with PT. Comorbidities, age, sex, obesity (BMI > 30 kg/m^2^), chronic therapy including anticoagulation and antiaggregation therapy were not associated with PT in our cohort.

However, on multivariable logistic regression analysis, only D-dimer ≥ 1.0 mg/L (OR 1.78, 95% CI 1.12–2.75) and high-flow supplemental oxygen (≥15 L/min) (OR 1.89, 95%CI 1.26–2.84) were independently associated with PT.

Among patients that had both factors present, elevated D-dimer and high oxygen need, 49% (56 of 115) had PT (calculated sensitivity of 32% and specificity of 88% with positive predictive value of 49%, 95%CI 39–57%). PT had 31% (90 of 292) of patients who had only one risk present, and 19% (32 of 168) without any. Frequencies of PT stratified by oxygen support and D-dimer levels are presented in Table 4.

Then, we performed a separate analysis that excluded patients with subsegmental PT since the clinical significance of this finding is unclear. In a multivariable analysis, male sex (OR 0.52, 95%CI 0.31–0.85), D-dimer ≥ 1.0 mg/L (OR 2.45, 95%CI 1.47–4.14), and O_2_ ≥ 15 L (OR 2.58, 95%CI 1.47–3.45) were associated with the development of non-subsegmental PT.

Furthermore, we separately examined factors associated with PT in patients requiring low-flow supplemental oxygen. On multivariable analysis, only D-dimer ≥ 1.0 mg/L was associated with PT (OR 1.86, 95%CI 1.03–3.28), even when subsegmental thrombosis was excluded from the analysis (OR 3.49, 95%CI 1.63–7.94).

### 3.4. Factors Associated with Mortality

Finally, we examined the factors associated with mortality. During hospital stay, 40 patients died, and 32.5% of them had PT. On univariable analysis patients who died were significantly older (65, IQR 58–74 vs. 61, IQR 50–69 years, *p* = 0.0170) and more frequently had arterial hypertension (28, 70% vs. 240, 44.86%, *p* = 0.0027, OR 2.86, 95%CI 1.44–5.53). There were no differences in other comorbidities. Regarding laboratory findings on admission, only serum fibrinogen levels (5.5, IQR 4.5–6.3 vs. 6.0, IQR 5.2–7.2, *p* = 0.0295) and platelets (155, IQR 129–216 vs. 192, IQR 144/256, *p* = 0.0179) were significantly decreased in patients who died during hospitalization.

On multivariable analysis, age ≥ 65 years (OR 3.20 (95%CI 1.30–8.42, *p* = 0.0138), high-flow supplemental oxygen ≥ 15 L/min (OR 7.33, 95%CI 1.70–50.33, *p* = 0.0151) and ICU admission (OR 35.22, 95%CI 13.63–107.0, *p* < 0.0001) were linked with mortality. We found no association between in-hospital mortality and PT.

## 4. Discussion

In our cohort study, we found a high incidence of PT (30.9%) in hospitalized COVID-19 patients mainly affecting subsegmental and segmental pulmonary arterial branches (83.1%). Additionally, we noted a significant correlation between D-dimer levels, COVID-19 severity and PT.

There is significant heterogeneity among studies reporting PT incidence in COVID-19 patients, which might be due to the different protocols used or the availability of CTPA. In the Italian cohort study, CTPA was systematically performed in hospitalized patients (*n* = 333) and PT was diagnosed in 33% of cases [17]. Similarly, a retrospective study from Spain found PT in 30% of patients undergoing CTPA (*n* = 242) [18]. Kollias et al. report the same incidence in their meta-analysis as the Spanish cohort [19]. On the contrary, a large French cohort study (*n* = 1240) reported the incidence of 8.3% which could be, as the authors suggest, due to a large number of non-severe patients [20].

Currently, there are no consensus guidelines concerning which patients should undergo CTPA. Several studies suggest that patients should be screened for PT when presenting with otherwise unexplained respiratory deterioration, dyspnea, hemoptysis, or high D-dimer levels [21]. However, these signs are unspecific and frequently accompany severe forms of COVID-19. Current guidelines recommend that LMWH should be used as part of treatment in all hospitalized patients, in therapeutic doses for patients outside of ICU, and prophylactic dosage for the critically ill [22]. Those are based on the REMAP-CAP trial data that showed benefit on mortality of therapeutic dose of LMWH in patients with moderate severity of COVID-19, whereas the benefit was not shown for critically ill patients [23,24]. However, these trials did not provide a protocol on performing a CTPA. The question remains, are we missing patients with undiagnosed PT who might benefit from therapeutic doses and prolonged anticoagulation.

It appears that traditional risk factors for PT, such as age, previous VTE, or history of malignant disease differ from those associated with COVID-19 [3]. Male sex, longer delay from symptom onset to hospitalization, and the degree of immune response were associated with PT development in COVID-19 patients [20]. Similarly, we found that patients with PT had higher levels of LDH and CRP suggesting a more severe inflammatory response. We also examined the association of neutrophil-to-lymphocyte and platelet-to-lymphocyte ratio with COVID-19 severity and PT diagnosis but found no correlations, in contrast to other studies that found significant associations with these hematological markers and thromboinflammation [25,26]. This supports the hypothesis of immunothrombosis as a probable pathogenetic mechanism, which was further shown on histopathological changes in lungs in COVID-19 patients [5,12,27].

Furthermore, we found that patients who required high-flow supplemental oxygen (≥15 L/min, HFNC) or more aggressive respiratory support (NIV, IMV and ECMO) had the highest incidence of PT. Bompard F. et al. also reported a higher rate of mechanical ventilation and ICU admission in patients with PT [28]. In contrast, a more recent Italian observational study found no difference in ICU admission, requirements for CPAP, or intubation [29].

D-dimer value serves as a useful indicator reflecting blood clot formation. Several studies showed that D-dimer are elevated in COVID-19 patients with PT, however, the reported cut-off values varied, and the optimal threshold should be higher than the traditional cutoff value (<0.5 mg/L) used as VTE exclusion criteria [12,17,18,30].

We have also found a high prevalence of subsegmental PT, especially in low-flow supplemental oxygen group with D-dimer < 1.0 mg/L. In this group, 53% of the PT diagnosed were subsegmental, in contrast to 30% in high-flow supplemental oxygen group with D-dimer ≥ 1.0 mg/L. The clinical impact of the incidental finding of subsegmental PT on CTPA and its possible association with COVID-19 complications including long-COVID have not been extensively studied. A high prevalence of subsegmental PT should probably be expected considering predominant peripheral and subpleural distribution of inflammatory lesions in COVID-19 and endothelitis as a proposed pathogenic mechanism of PT [5,31].

One could argue that in moderately ill patients with clinically unsuspected PT, routine CTPA could lead to the overdiagnosis of isolated subsegmental PT and more patients could be exposed to unnecessary anticoagulation with possible complications. However, at this point, we do not know which patients from this subgroup could benefit from treatment.

The impact of PT on COVID-19 mortality is unclear. A Brazilian cohort (*n* = 4120) reported twice as high in-hospital mortality in patients diagnosed with VTE. In contrast, we found no association between PT and mortality, similar as others [17,32]. This might be explained by a high frequency of small peripheral thrombi, which might not cause further respiratory deterioration.

Our study should be viewed in light of its limitations; this was a monocentric study; CTPA was not performed routinely on all consequently hospitalized patients with COVID-19 and the indication for CTPA was left at physicians’ discretion, which could lead to selection bias; ultrasound screening for deep vein thrombosis in lower extremities was not performed, even though DVT is not recognized as a risk factor in patients with PT in COVID-19 [10]. Additionally, the same CTPA images were not revised by different radiologists. The incidence of PT in critically ill patients might be underestimated since most of them did not undergo CTPA due to the UHID ICU organization at that time. The impact of nosocomial infections on mortality was not examined. There was no long-term follow-up, so the clinical impact of PT could not be fully evaluated. The study was performed when the Delta variant of COVID-19 was predominant, so these results do not necessarily need to extrapolate in the Omicron era. Nevertheless, we included a relatively large and well-defined cohort of patients and showed an association of PT with disease severity and D-dimer rather than with traditional risk factors.

In conclusion, according to our data, it seems reasonable to perform CTPA in all patients requiring high-flow supplemental oxygen (≥15 L/min), especially if they also have D-dimer ≥ 1.0 mg/L. In this group, 49% had PT. For patients requiring low-flow supplemental oxygen, D-dimer levels and clinical suspicion can suggest the need for CTPA. Neither peripheral nor central PT were associated with mortality. The clinical significance of PT and its impact on outcomes remains unclear, and further studies are needed to resolve this clinical puzzle.

## Figures and Tables

**Figure 1 jcm-11-07049-f001:**
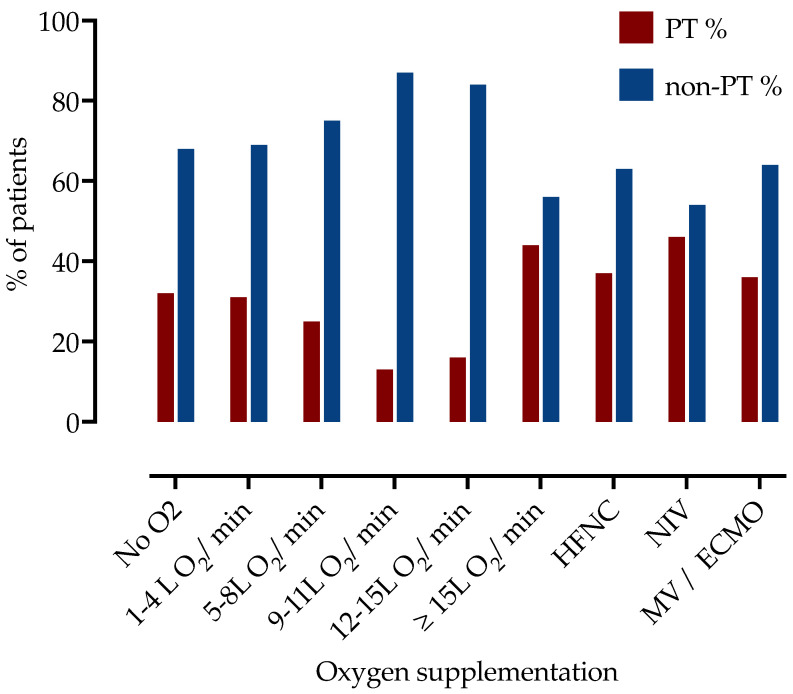
Frequency of patients with PT stratified by the respiratory support level. PT—pulmonary thrombosis, HFNC—high flow nasal cannula, NIV—non-invasive ventilation, IMV—invasive mechanic ventilation, ECMO—extracorporeal membrane oxygenation.

**Table 1 jcm-11-07049-t001:** Baseline patients’ characteristics.

	Pulmonary Thrombosis (*n* = 178)	No Pulmonary Thrombosis (*n* = 397)	*p*-Value ^a^
Age, median (IQR) ^b^	61 (50–69)	62 (51–70)	0.6400
Male, No. (%) ^b^	124 (69.7)	258 (65.0)	0.2941
**Comorbidities**			
Diabetes Mellitus	37 (20.8)	66 (16.6)	0.2406
Arterial Hypertension	83 (46.6)	185 (46.6)	>0.9999
Gastritis/GERD	8 (4.5)	20 (5.0)	>0.9999
Cardiovascular Disease	37 (20.8)	107 (27.0)	0.1197
**Clinical findings on admission**			
Duration of illness on admission, days	10 (8–13)	10 (8–12)	0.1906
Peripheral oxygen saturation (spO_2_)	91 (88–94)	92 (88–94)	0.1384
MAP	97 (90–106)	97 (88–102)	0.0549
CHOD score	5 (3–7)	5 (3–5)	**0.0031**
**Laboratory findings on admission**			
C-reactive Protein, mg/L	107 (54–177)	83 (47–148)	**0.0128**
Procalcitonin, µg/L	0.15 (0.082–0.32)	0.13 (0.084–0.27)	0.6218
Interleukin 6, ng/L	66 (31–108)	59 (32–121)	0.8638
Ferritin, µg/L	884 (536–1511)	1042 (564–2018)	0.3808
White Blood Cells Count, ×10^9^/L	7.8 (5.3–11.0)	6.8 (5.2–9.8)	0.0517
Lymphocyte count, 10^9^/L	0.5 (0.1–0.7)	0.7 (0.5–0.9)	0.1559
Neutrophil/lymphocyte ratio	8.8 (4.9–13.0)	7.9 (5.1–13)	0.8200
Hemoglobin, g/L	137 (129–145)	137 (128–147)	0.4048
Platelets, ×10^9^/L	201 (149–274)	181 (141–245)	**0.0131**
Platelets/lymphocyte ratio	283 (187–424)	260 (180–425)	0.5877
Aspartate Aminotransferase, IU/L	52 (36–75)	52 (38–80)	0.4993
Alanine Aminotransferase, IU/L	45 (28–66)	44 (27–72)	0.8851
Gamma-glutamyl transferase, IU/L	55 (34–94)	52 (32–99)	0.7702
Lactate dehydrogenase, IU/L	410 (324–519)	380 (294–478)	**0.0266**
Prothrombin time	1.1 (0.96–1.2)	1.1 (1.0–1.2)	**0.0030**
International normalized ratio	0.97 (0.93–1.1)	0.95 (0.91–1.0)	**0.0026**
Fibrinogen, g/L	6 (5.1–7.2)	5.9 (5.2–7.1)	0.9713
D-dimer, mg/L	1.2 (0.74–3)	0.94 (0.59–1.5)	**<0.0001**

^a^—Fisher exact or Mann–Whitney U test, as appropriate. ^b^—data are presented as medians (interquartile range) or frequencies (percentage). Abbreviations: IQR—interquartile range, GERD—gastroesophageal reflux disease, SpO_2_—peripheral oxygen saturation, MAP—mean arterial pressure, CHOD (CRP concentration, hearth rate, oxygen saturation, D-dimer).

**Table 2 jcm-11-07049-t002:** Frequency of subsegmental and combined segmental, lobar, or central PT stratified by oxygen supplementation requirement in hospitalized patients with COVID-19.

Oxygen Supplementation	Number of Patients	Subsegmental PT	Segmental, Lobar, or Central PT	No PT
**No oxygen**	44	6 (13.6%)	8 (18.2%)	30 (68.2%)
**1–4 L O_2_/min**	128	17 (13.3%)	23 (18.0%)	88 (68.8%)
**5–8 L O_2_/min**	65	5 (7.7%)	11 (16.9%)	49 (75.4%)
**9–11 L O_2_/min**	38	1 (2.6%)	4 (10.5%)	33 (86.8%)
**12–15 L O_2_/min**	69	6 (8.7%)	5 (7.2%)	58 (84.1%)
**≥15 L O_2_/min**	77	10 (13.0%)	24 (31.2%)	43 (55.8%)
**HFNC**	92	13 (14.1%)	21 (22.8%)	58 (63.0%)
**NIV**	24	4 (16.7%)	7 (29.2%)	13 (54.2%)
**IMV and ECMO**	44	3 (6.8%)	12 (27.3%)	29 (65.9%)

Data are presented as frequencies with percentages. Chi-square test showed significant difference between group (*p* = 0.0018). PT—pulmonary thrombosis, HFNC—high flow nasal cannula, NIV—non-invasive ventilation, IMV—invasive mechanic ventilation, ECMO—extracorporeal membrane oxygenation.

**Table 3 jcm-11-07049-t003:** Univariable and multivariable analysis of factors associated with pulmonary thrombosis.

	Univariable Analysis		Multivariable Analysis	
	OR (95% CI)	*p*-Value	OR (95% CI)	*p*-Value
Age	0.89 (0.62–1.28)	0.5804		
Male sex	1.24 (0.84–1.79)	0.2941		
CHOD ≥ 5	1.64 (1.14–2.38)	0.0085		
Plt ≥ 185 ×10^9^/L	1.46 (1.02–2.09)	0.0381		
CRP ≥ 85 mg/L	1.52 (1.06–2.17)	0.0240		
D-dimer ≥ 1.0 mg/L	1.87 (1.29–2.69)	0.0006	1.78 (1.12–2.75)	0.0082
O_2_ ≥ 15 L	2.03 (1.43–2.89)	0.0001	1.89 (1.26–2.84)	0.0021
ICU admission	1.81 (1.07–3.09)	0.0320		

The strength of association was expressed as odds ratios (OR) and its corresponding 95% confidence interval (CI). The area under the ROC curve in the fully adjusted model was AUC 0.64 (95% CI 0.59 to 0.69). CHOD (CRP concentration, hearth rate, oxygen saturation, D-dimer); CRP—C-reactive protein, ICU—intensive care unit.

**Table 4 jcm-11-07049-t004:** Frequencies of PT stratified by oxygen support and D-dimer levels.

	Low-Flow Oxygen Supplementation (<15 L/min)		High-Flow Oxygen Supplementation (≥15 L/min)	
	D-dimer < 1.0 mg/L (*n* = 167)	D-dimer ≥ 1.0 mg/L (*n* = 179)	D-dimer < 1.0 mg/L (*n* = 114)	D-dimer ≥ 1.0 mg/L (*n* = 115)
PT incidence	32 (19.1%)	54 (30.2%)	36 (31.6%)	56 (48.7%)
Subsegmental PT	17 (10.2%)	18 (10.1%)	12 (10.5%)	17 (14.8%)
Segmental PT	11 (6.6%)	26 (14.5%)	19 (16.7%)	28 (24.3%)
Lobar PT	3 (1.8%)	7 (3.9%)	5 (4.4%)	7 (6.1%)
Central PT	1 (0.6%)	3 (1.7%)	0	4 (3.5%)

PT—pulmonary thrombosis.

## Data Availability

The datasets generated and/or analyzed during the current study are available from the corresponding author on reasonable request.

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
