# Peer review of "COVID-19 and Pulmonary Thrombosis—An Unresolved Clinical Puzzle: A Single-Center Cohort Study"

_jcm, 2022, doi:10.3390/jcm11237049_

Round 1
Reviewer 1 Report
1. Line 36- these pathogenic factors seem to play role, at least partially, in thrombotic events in patients with celiac disease and this should be added ( https://www.ncbi.nlm.nih.gov/pmc/articles/PMC9144428/)
2. How many patient had secondary bacterial pneumonia in this cohort?
3. Were all patients treated with LMWH? How about patients who were already on anticoagulation with DOACs or patients with advanced CKD where LMWH would be contraindicated?
4. In methodology CHOD score calculation should be explained- this is an international journal and not all countries are using this scoring system. particularly in North America
5. Table 1- platelet and INR were significantly different in the two group. Since both values are affected in people with cirrhosis have you controlled for this co-morbidity?
6. Factors associated with morality- we do not know about the burden of medical co-morbidities in these two groups so I am not convinced that this is accurate result and conclusion
7. Have you controlled for BMI in the two groups? It is recognized that obesity is a risk factors for both covid and VTE so it would be intuitive to control for this variable
Author Response
- Line 36- these pathogenic factors seem to play role, at least partially, in thrombotic events in patients with celiac disease and this should be added (https://www.ncbi.nlm.nih.gov/pmc/articles/PMC9144428/)
ANSWER: We thank the reviewer for suggesting an interesting article. Similar pathogenetic role have been suggested in a variety of conditions. However, we believe that adding the data on the role of thrombosis and hypercoagulability in celiac disease would be confusing and doesn’t fit in the concept of manuscript.
- How many patients had secondary bacterial pneumonia in this cohort?
ANSWER: None of the patients included on admission had secondary bacterial pneumonia. The impact of nosocomial infections was not analyzed.
This was added to the manuscript limitations: “The impact of nosocomial infections on mortality was not be examined.” Line 274-275
- Were all patients treated with LMWH? How about patients who were already on anticoagulation with DOACs or patients with advanced CKD where LMWH would be contraindicated?
ANSEWR: We thank the reviewer for noticing that we did not include data on previous anticoagulation. Overall, 22 patients from PT and 52 from non-PT group received either acetylsalicylic acid (15 and 41), warfarin (1 and 7) or DOACs (5 vs 5) and one patient from PT group was on fraxiparine before hospitalization. As per hospital protocol, and international recommendations at that time, at admission DOACs were converted to LMWH and dosed according to weight and renal function.
This is now added in the manuscript:
In Methods:
“As per hospital protocol and international guidelines at that time, direct oral anticoag-ulants and warfarine were converted to LMWH dosed according to body weight and renal function.” (Line 67-69)
Results: “In addition, there was no difference in chronic anticoagulation or antiagregation (acetylsalicylic acid n=56; warfarin n=7; rivaroxaban n=7; apixaban 3; fraxiparine n=1).”(Line 127-129)
In addition, previous DOACs or antiagregation therapy was not associated with PT in our cohort.
“Comorbidities, age, sex, obesity (BMI>30kg/m2), chronic therapy including anticoagulation and antiagregation therapy were not associated with PT in our cohort.” (Line 166-168)
- In methodology CHOD score calculation should be explained- this is an international journal and not all countries are using this scoring system. particularly in North America
ANSWER: We added the definition of CHOD score.
“CHOD, an acronym of (CRP concentration + Heart rate + Oxygen saturation + D-Dimer levels) was calculated for each patient, as previously reported (REF – PMID 33440207). (Line 61-63)
- Table 1- platelet and INR were significantly different in the two group. Since both values are affected in people with cirrhosis have you controlled for this co-morbidity?
ANSWER: Patients with PT had significantly higher both Plt, and PT of 97% (INR 1.1) so this is probably not attributed to decompensated liver disease, since in both groups these values were within reference range. This was an incidental laboratory finding that is hard to clinically explain. Regarding liver cirrhosis, only 2 patients in our cohort had liver cirrhosis.
- Factors associated with morality- we do not know about the burden of medical co-morbidities in these two groups so I am not convinced that this is accurate result and conclusion
ANSWER: We thank the reviewer for that comment – in the original version of manuscript we presented only the data from multivariable analysis. In univariable analysis patients who died were significantly older (65, IQR 58-74 vs 61, IQR 50-69, p=0.0170) and more frequently had arterial hypertension (28, 70% vs 240, 44.86%, p=0.0027, OR 2.86, 95%CI 1.44-5.53). There were no differences in other comorbidities. Regarding laboratory findings on admission, only serum fibrinogen levels (5.5, IQR 4.5-6.3 vs 6.0, IQR 5.2-7.2, p=0.0295) and platelets (155, IQR 129-216 vs 192, IQR 144/256, p=0.0179) were significantly decreased in patients who died during hospitalization. This is now added in the manuscript – line 194-200.
- Have you controlled for BMI in the two groups? It is recognized that obesity is a risk factors for both covid and VTE so it would be intuitive to control for this variable
ANSWER: Yes, obesity nor other comorbidities were associated with PT as stated above – line 166-168.
Reviewer 2 Report
Dear Editor,
I have read the paper by Loris Močibob et al with interest. The manuscript addresses the evaluate the clinical and laboratory characteristics associated with PT in patients with COVID-19.
The Study found that is reasonable to perform CTPA in all pa- 278 tients requiring high-flow supplemental oxygen (³ 15L/min), especially if they also have 279 D-dimer ³ 1.0 mg/L. In this group 49% of them had PT. For patients requiring low-flow 280 supplemental oxygen, D-dimer levels and clinical suspicion can suggest the need for 281 CTPA. Neither peripheral nor central PT were associated with mortality.
In general, this paper is well-written and requires substantial revision. Several sentences are short, hard to read or understand, and often statements are not supported by proper referencing. Furthermore, I have encountered several formatting issues, and typos.
In the “Introduction” the authors should emphasize the negative impact of the COVID-19 Pandemic on medical activity worldwide. I suggest citing the following reference:
· https://doi.org/10.3390/jcm11092452
· https://doi.org/10.3390/medicina58091197
· https://doi.org/10.3390/medicina58101322
· https://doi.org/10.3390/cancers14174338
In the “Results” Considering that the authors have the information regarding the red blood cell (lymphocyte, and platelets as wee seen in Table 1) it will be interesting to compare at ROC analysis and multivariate analysis the predictive role of Neutrophil-to-Lymphocyte ratio and Platelet-to-Lymphocyte ratio between the PT and non-PT group. Furthermore, they reported in manuscript numerously systemic inflammatory markers such CRP, Procalcitonin, IL-6, Ferritin, Fibrinogen, and D-Dimer, but only CRP, and D-Dimer was statically significant (p=0.01 and p<0.0001). It will be interesting to compare the statistically signific value of well-known inflammatory biomarkers with hematological ratios
- https://doi.org/10.3390/diagnostics12092089
The Discussion section needs to be improved with multiple other articles and to compare the results with other inflammatory markers value of COVID-19 and non-COVID-19 thromboembolic events. Moreover, recently in literature were published highly quality paper with interesting result regarding the hematological ratios and poor outcome of COVID-19 patients. Moreover, the hematological ratios are an independent predictor of thromboembolic events in COVID-19 and non-COVID-19 patients. I suggest the following reference:
- https://doi.org/10.3390/antibiotics11010060
- https://doi.org/10.3390/jcm10194343
- https://doi.org/10.3390/jcm10184058
Kind regards,
Author Response
- In the “Introduction” the authors should emphasize the negative impact of the COVID-19 Pandemic on medical activity worldwide. I suggest citing the following reference:
- https://doi.org/10.3390/jcm11092452
- https://doi.org/10.3390/medicina58091197
- https://doi.org/10.3390/medicina58101322
- https://doi.org/10.3390/cancers14174338
ANSWER: We thank the reviewer for suggesting these interesting articles. However, we believe that adding this information doesn’t really fit in the concept of our manuscript.
- In the “Results” Considering that the authors have the information regarding the red blood cell (lymphocyte, and platelets as wee seen in Table 1) it will be interesting to compare at ROC analysis and multivariate analysis the predictive role of Neutrophil-to-Lymphocyte ratio and Platelet-to-Lymphocyte ratio between the PT and non-PT group. Furthermore, they reported in manuscript numerously systemic inflammatory markers such CRP, Procalcitonin, IL-6, Ferritin, Fibrinogen, and D-Dimer, but only CRP, and D-Dimer was statically significant (p=0.01 and p<0.0001). It will be interesting to compare the statistically signific value of well-known inflammatory biomarkers with hematological ratios
- https://doi.org/10.3390/diagnostics12092089
ANSWER: We thank the reviewer for the suggestions and for recommending an interesting article. We have calculated and added the values for Neutrophil-to-Lymphocyte (NLR) and Platelet-to-Lymphocyte ratio (PLR) to Table 1. However, we have found no statistically significant difference in our cohort. This is now added in the results:
“… including the platelet-to-lymphocyte and neutrophil-to-lymphocyte ratio. Line 124-125
Furthermore, we preformed the suggested ROC analysis and found no significant difference for NLR (AUC 0.5059, p=0.8198) and PLR (AUC 0.5141, p=0.5875). We did not perform multivariate analysis of the aforementioned ratios since no significant difference was found in univariant analysis. Correlation analysis was preformed, as advised by the reviewer. We found that ANC/ALC positively correlates with CRP, PCT, fibrinogen, and D-dimer values, while PLT/ALC correlates with CRP, ANC/ALC, fibrinogen, and D-dimer levels. However, we did not find any correlation between these ratios and pulmonary thrombosis which was the main aim of our study. We believe that including this analysis in manuscript would exceed the aims of our study and might be confusing to readers.
- The Discussion section needs to be improved with multiple other articles and to compare the results with other inflammatory markers value of COVID-19 and non-COVID-19 thromboembolic events. Moreover, recently in literature were published highly quality paper with interesting result regarding the hematological ratios and poor outcome of COVID-19 patients. Moreover, the hematological ratios are an independent predictor of thromboembolic events in COVID-19 and non-COVID-19 patients. I suggest the following reference:
- https://doi.org/10.3390/antibiotics1101006
- - https://doi.org/10.3390/jcm1019434
- - https://doi.org/10.3390/jcm10184058
ANSWER: We thank the reviewer for that comment.
We added following to the discussion: „We also examined the association of neutrophil-to-lymphocyte and platelet-to-lymphocyte ratio with COVID-19 severity and PT diagnosis, but found no correlations. in contrast to other studies that found significant associations with these hematological markers and thromboinflammation [25,26]. Line 233-237
Reviewer 3 Report
Croatian researchers conducted an attractive and relatively well-planned study on thrombotic complications in COVID-19. This topic is indeed known, but considering the importance of the clinical problem of these complications, especially in severe COVID-19, the study is valuable.
I have attached notes to the manuscript below:
1. I ask the authors to consider changing the name pulmonary thrombosis to pulmonary embolism. The latter term is used more often.
2. I agree with the authors that the high risk of PE in COVID-19 is due to hypercoagulability and a cytokine storm. However, the condition may also result from occult hyperfibrinolysis. Please add this information to the introduction and the literature (DOI: 10.1055/a-1346-3178).
3. Materials and methods - I have no significant comments; this chapter is carefully written. A question for the authors - has the same radiologist always examined the included patients? If not, please consider it as a limitation of the study.
4. Results - clearly presented. I have no critical comments.
5. Discussion - clear and lucid, relates to the results of own research.
Overall, well a written manuscript worth publishing.
Author Response
Croatian researchers conducted an attractive and relatively well-planned study on thrombotic complications in COVID-19. This topic is indeed known, but considering the importance of the clinical problem of these complications, especially in severe COVID-19, the study is valuable.
I have attached notes to the manuscript below:
- I ask the authors to consider changing the name pulmonary thrombosis to pulmonary embolism. The latter term is used more often.
ANSWER: We thank the reviewer for that comment. Since our results support the immunothrombosis hypothesis and formation of thrombi in-situ, we decided to use the term pulmonary thrombosis to highlight different pathophysiology, clinical presentations and risk factors than pulmonary embolism in other conditions.
- I agree with the authors that the high risk of PE in COVID-19 is due to hypercoagulability and a cytokine storm. However, the condition may also result from occult hyperfibrinolysis. Please add this information to the introduction and the literature (DOI: 10.1055/a-1346-3178).
ANSWER: We thank the reviewer for making that point. We have added this as a possible mechanism and cited suggested paper. “In situ PT may be caused by direct endothelial injury due to local inflammation and hypercoagulable state in combination with occult hyperfibrinolysis, as was recently reported [4,5,6].“ Line 35-37.
- Materials and methods - I have no significant comments; this chapter is carefully written. A question for the authors - has the same radiologist always examined the included patients? If not, please consider it as a limitation of the study.
ANSWER: The CTPA was examined by three experienced radiologists. We have added this in the manuscript.
Methods: “A total of three experienced senior radiology consultants were included in image evaluation with one CTPA scan being analyzed by one radiologist.” Line 101-103
Discussion: “Additionally, same CTPA images were not revised by different radiologists.” Line 272-273
- Results - clearly presented. I have no critical comments.
ANSWER: We thank the reviewer for nice comments.
- Discussion - clear and lucid, relates to the results of own research.
ANSWER: We thank the reviewer for nice comments.
Round 2
Reviewer 2 Report
no further comments
Reviewer 3 Report
The authors have addressed all my concerns.
Author Response
We thank the reviewer.